# Protein Supplementation May Dampen Positive Effects of Exercise on Glucose Homeostasis: A Pilot Weight Loss Intervention

**DOI:** 10.3390/nu15234947

**Published:** 2023-11-29

**Authors:** John A. Batsis, Dakota J. Batchek, Curtis L. Petersen, Danae C. Gross, David H. Lynch, Hillary B. Spangler, Summer B. Cook

**Affiliations:** 1Division of Geriatric Medicine, University of North Carolina at Chapel Hill School of Medicine, Chapel Hill, NC 27599, USA; david_lynch@med.unc.edu (D.H.L.); hillary.spangler@unchealth.unc.edu (H.B.S.); 2Department of Nutrition, Gillings School of Global Public Health, University of North Carolina at Chapel Hill, Chapel Hill, NC 27599, USA; dbatchek@unc.edu (D.J.B.); dcgross@unc.edu (D.C.G.); 3The Dartmouth Institute for Health Policy, Dartmouth College, Hanover, NH 03755, USA; clpurtis@gmail.com; 4Center for Aging and Health, University of North Carolina at Chapel Hill, Chapel Hill, NC 27599, USA; 5Geisel School of Medicine at Dartmouth, Dartmouth College, Hanover, NH 03755, USA; 6Department of Kinesiology, University of New Hampshire, Durham, NH 03824, USA; summer.cook@unh.edu

**Keywords:** protein supplementation, exercise, resistance training, weight loss, glucose metabolism, glucose homeostasis, insulin resistance, inflammation, older adults

## Abstract

Background: The role of protein in glucose homeostasis has demonstrated conflicting results. However, little research exists on its impact following weight loss. This study examined the impact of protein supplementation on glucose homeostasis in older adults >65 years with obesity seeking to lose weight. Methods: A 12-week, nonrandomized, parallel group intervention of protein (PG) and nonprotein (NPG) arms for 28 older rural adults (body mass index (BMI) ≥ 30 kg/m^2^) was conducted at a community aging center. Both groups received twice weekly physical therapist-led group strength training classes. The PG consumed a whey protein supplement three times per week, post-strength training. Primary outcomes included pre/post-fasting glucose, insulin, inflammatory markers, and homeostasis model assessment of insulin resistance (HOMA-IR). Results: Mean age and baseline BMI were 72.9 ± 4.4 years and 37.6 ± 6.9 kg/m^2^ in the PG and 73.0 ± 6.3 and 36.6 ± 5.5 kg/m^2^ in the NPG, respectively. Mean weight loss was −3.45 ± 2.86 kg in the PG and −5.79 ± 3.08 kg in the NPG (*p* < 0.001). There was a smaller decrease in pre- vs. post-fasting glucose levels (PG: −4 mg ± 13.9 vs. NPG: −12.2 ± 25.8 mg/dL; *p* = 0.10), insulin (−7.92 ± 28.08 vs. −46.7 ± 60.8 pmol/L; *p* = 0.01), and HOMA-IR (−0.18 ± 0.64 vs. −1.08 ± 1.50; *p* = 0.02) in the PG compared to the NPG. Conclusions: Protein supplementation during weight loss demonstrated a smaller decrease in insulin resistance compared to the NPG, suggesting protein may potentially mitigate beneficial effects of exercise on glucose homeostasis.

## 1. Introduction

Obesity is a condition characterized by a body mass index (BMI) ≥ 30 kg/m^2^ and has well-established associations with various chronic health conditions such as type 2 diabetes (T2D) and cardiovascular disease [1]. Glucose homeostasis describes the maintenance of proper blood glucose levels via the counteracting effects of peptide hormones insulin and glucagon [2]. Insulin resistance—a disruption in glucose homeostasis—is strongly associated with the development of T2D, abdominal obesity, and key components of the metabolic syndrome, all of which have a tendency to be proinflammatory [3,4]. Weight loss due to energy restriction may be associated with improvements in insulin resistance and metabolic syndrome that subsequently can reduce inflammatory burden and oxidative stress [5]. Additionally, energy-restricted weight loss has also been shown to reduce levels of cholesterol, decreasing the risk of developing cardiovascular disease [6]. Protein is known to induce satiety, which may facilitate adherence to a moderate energy restriction diet (e.g., ~500 kcal) during weight loss efforts. However, protein has also been shown to blunt an intervention’s effect on weight loss [7]. Specifically, when energy intake is fixed, increased protein consumption mitigates the caloric deficit and may be associated with less than favorable outcomes [8].

Past research has focused more on different levels of exercise intensity on glucose homeostasis [9], as well as the general effects of ageing on glucose homeostasis [10]. Physical therapist-lead strength training sessions were found to improve physical function in older adults, as significant improvements were seen in 6 min walk, gait speed, and 5 times sit-to-stand times [11]. Caloric restriction is also known to independently have a favorable impact on glucose homeostasis, by reducing insulin resistance and improving glucose uptake [12]. Further, current research has begun looking at nuances associated with time-restricted eating, weight loss, and metabolic health [13,14]. Time-restricted eating, a form of intermittent fasting, has been speculated to provide no additional improvement in body composition or cardiometabolic health [13]. There has also been recent interest in probiotic supplementation related to the gut microbiome. Although the findings demonstrated no effect on body composition, the supplement did promote growth in gut bacteria associated with positive health outcomes [15]. Other research focusing on body composition and glucose metabolism have found weight loss to be associated with improved glycemic control [16]. For instance, lower fasting insulin was observed among those who maintained weight loss following an intervention compared to those who relapsed [16].

It is well established that adults with obesity may also have elevated levels of systemic inflammation [17], and that weight loss interventions decrease circulating inflammatory markers such as interleukin-6 (IL-6) and tumor necrosis factor-alpha (TNF-α) [17,18,19,20]. However, research in subgroups with T2D is less concrete, as there are alternate perspectives to the roles of such inflammatory markers among patients with and without T2D whose physiology underlies that of impaired glucose homeostasis [21]. Lower levels of circulating IL-6 among persons with T2D has been observed compared to controls, suggesting IL-6 may play an anti-inflammatory role, and that persons with T2D have a poorer inflammatory response to tissue damage and infection [21]. Another study reproduced this finding while also noting that inflammatory markers TNF-α and IL-1β were not significantly greater among patients with T2D [21,22]. This suggests that patients with T2D may have adapted to chronic exposure to hyperglycemia, perhaps explaining the nonsignificant difference in inflammatory marker levels among patients with and without T2D [22]. In summary, heterogeneous findings regarding obesity, impaired glucose homeostasis, and inflammatory markers illustrate research gaps that should continue being addressed in future studies.

As for weight change, there are concerns that weight loss can potentially lead to a reduction in muscle mass and function, termed sarcopenia [23,24]. This is of importance, as ageing also decreases an individual’s ability to synthesize dietary protein in response to resistance exercise, termed anabolic resistance [25,26], which may exacerbate muscle loss [25,27]. Protein and amino acid supplementation are often a recommended means to enhance protein intake to mitigate anabolic resistance [25]. Specifically, studies suggest that ingesting a protein bolus immediately after resistance exercise upregulates muscle protein synthesis (MPS) in older adults through mammalian target of rapamycin complex 1 (mTORC1) pathway activation [28]. However, hyperactivation of mTORC1 from chronically high levels of protein intake (greater than the RDA’s recommended 0.8 g/kg body weight (BW)) have been shown to induce insulin resistance [29,30]. Increased activity of mTORC1 induces insulin resistance by phosphorylating insulin receptor substrate 1 (IRS-1), thus inhibiting insulin-stimulated glucose uptake [31].

In contrast, other studies have demonstrated that high-protein diets improve insulin resistance and glycemic control [32,33]. One study found a high-protein diet more effective at reducing markers of insulin resistance compared to the Mediterranean diet [33]. A separate study highlighted how short-term high-protein diets improve prandial insulin secretion [32]. Furthermore, muscle contraction during exercise has been shown to improve glucose homeostasis via stimulation of the GLUT-4 glucose transporter, allowing for uptake of glucose from the bloodstream into the cell [34]. Such conflicting data for ways in which glucose homeostasis is worsened or enhanced via exercise and protein intake has created many questions. Few studies have been conducted that use a combined protein–exercise intervention to analyze elements of insulin physiology and glucose homeostasis. To begin answering these questions, as part of a previously conducted pilot feasibility study consisting of a weight loss intervention in older adults with obesity who were provided a whey protein supplement (protein group—PG) or not (nonprotein group—NPG), this study evaluated the combined intervention’s impact on glucose homeostasis.

## 2. Methods

### 2.1. Study Design

This study was a secondary analysis of a single-arm pilot study previously conducted in rural older adults with obesity (BMI ≥ 30 kg/m^2^). It was previously demonstrated that the group with protein supplementation (PG) had improved physical function despite a lower degree of weight loss [35]. A total of 28 participants were evaluated over 12 weeks in a community ageing center in northern New England. The full protocol, with recruitment, exercise, and dietary intervention details, has been previously described [35]. The Dartmouth-Hitchcock and University of North Carolina Institutional Review Boards approved the conduct of this study and all participants provided informed consent. This trial was registered on clinicaltrials.gov (NCT#03104192).

### 2.2. Exercise and Protein Intervention

Briefly, each study group (both n = 14) participated in a weekly nutrition session led by a registered dietician and a twice-weekly 75 min group exercise session led by a physical therapist. These exercise sessions were resistance-based and were performed at moderate intensity (13–15 rating on the Borg exertion scale) [11]. Daily aerobic exercise was performed with weekly physical therapist guidance and coaching as part of their regimen [11]. Each group of participants were part of separate studies that engaged in the same intervention other than the consumption of protein. Half of the participants (n = 14) consumed one serving of a whey protein supplement powder (100 kcal) of their choosing, constituted in cold water (Thorne Research, Dover, ID, USA) (27 g whey vanilla or 29 g whey chocolate; both of which with 2.2 g leucine per serving) three times per week. Additional constituents of the supplement included 1.9 g lysine, 1.2 g valine, and 1.4 g threonine. There was also 3 and 4 g of total carbohydrates, as well as 65 and 69 mg of calcium in the vanilla and chocolate powders, respectively. It should also be noted that the chocolate powder contained 200 mg of potassium whereas the vanilla powder contained only 70 mg. Two of the protein servings were consumed immediately after the exercise sessions, and one was self-administered at home. The whey supplement was dissolved in 8 oz of water and consumed within 30 min of completing the exercise regimen. As this was not a controlled feeding study, but an intervention study in the real world, participants did maintain some autonomy over their diet but were monitored and counseled by the registered dietician.

### 2.3. Data Collection

Body composition was assessed using the Seca medical body composition analyzer (mBCA) 514 bioelectrical impedance analyzer (BIA) (Hamburg, Germany). Self-reported physical activity data were entered into the BIA and participants stood on the apparatus without shoes or socks, holding the metal electrode with their knees slightly bent. Participant waist circumference and height were measured as previously described [35]. All participants had phlebotomy performed after an overnight fast by a trained phlebotomist at Dartmouth-Hitchcock using standard clinical procedures [35]. Assays were conducted either at the Dartmouth-Hitchcock laboratory or at the Mayo Clinic Laboratories via a send-out immediately after collection. Fasting plasma glucose was assessed using standard assays in a CLIA-certified laboratory, while interleukin-6 (IL-6), tumor necrosis factor-alpha (TNF-α), C-peptide, and insulin levels were evaluated using standard electrochemiluminescence immunoassays. Using the results of insulin, C-peptide, and fasting glucose levels, the homeostasis model assessment of insulin resistance (HOMA-IR) was determined. HOMA-IR is used to quantify insulin sensitivity [36], and its value was also used to determine β-cell function by quantifying changes in β-cell response to blood glucose levels [36]. HOMA-IR was first calculated by multiplying fasting plasma glucose and plasma insulin, then dividing by 22.4 (constant) (insulin × glucose)/22.5 [37]. These values were then compared to the homeostasis model assessment of β-cell function (HOMA-β-cell) index to determine β-cell function [38].

### 2.4. Statistical Analysis

All data were aggregated into a single dataset. A paired t-test was used to compare pre- and post-glucose homeostasis determinants within groups. An unpaired *t*-test with unequal variances and chi square were used to compare the change in values between PG and NPG groups for continuous values and categorical variables—both with and without controlling for age, sex, and weight loss. Effect sizes were conducted using the methods of Cohen and classified according to small (d = 0.2), medium (d = 0.5), and large (d ≥ 0.8) [35,39]. All analyses were conducted using R v3.6 (www.R-project.org). Statistical significance was defined as a *p*-value < 0.05 [35].

## 3. Results

Mean age was 72.9 ± 4.4 years in the PG and 73.0 ± 6.3 years in the NPG, respectively (*p* = 0.94) (Table 1). There were no significant differences in baseline characteristics between groups other than nonsmoking status (PG: 92.9 vs. NPG: 42.9%; *p* = 0.005). Table 2 represents the outcomes of the anthropometric and body composition measures. The PG had a smaller percent decrease in weight (−3.7 vs. −6.0%; *p* = 0.07), change in BMI (−1.36 vs. −2.15 kg/m^2^; *p* = 0.08), and change in fat mass index (−0.97 kg/m^2^ vs. −1.35 kg/m^2^; *p* = 0.37) compared to the NPG. Relative effect sizes favored the PG in fat-free mass index (d = +0.68) while favoring the NPG in percent weight change (d = −0.77), BMI (d = −0.71), appendicular lean mass (d = +0.30), fat mass index (d = −0.34), and visceral adipose tissue (d = −1.11).

Within both groups, there were decreases in fasting plasma glucose, insulin, C-peptide, insulin resistance and percent β-cell function (Table 3). However, the NPG showed a significantly greater unadjusted reduction in insulin level (d = −0.82, *p* = 0.01) and insulin resistance (d = −0.78, *p* = 0.02); these reductions had large and medium effect sizes in favor of the NPG, respectively. There was also a nonsignificant, but larger reduction in C-peptide (d = −0.66, *p* = 0.06) among the NPG. As for inflammatory markers, nonsignificant differences were observed within or between the PG and NPG. The only exception was a significantly greater reduction in TNF-α among the NPG (−0.19 pg/mL; *p* = 0.01, d = 0.45). Effect sizes for C-reactive protein (CRP) (d = −0.44) and erythrocyte sedimentation rate (ESR) (d = −0.31) low-moderately favored the NPG. Only IL-6 showed an effect size favoring the PG and was low in magnitude (d = +0.31) (Table 4). 

## 4. Discussion

Overall, consumption of protein supplementation post-exercise led to less of an impact on glucose homeostasis and inflammatory mediators in older adults with obesity participating in a weight loss intervention. While this novel pilot study provides provocative findings for an at-risk population, the results should be interpreted with caution and implications. 

This study’s preliminary findings align with previously conducted studies [40,41]. Specifically, lifestyle interventions are associated with enhanced glycemic control and insulin sensitivity as observed in persons with obesity and T2D [40]. However, higher protein intake has been shown to reduce insulin sensitivity while concomitantly increasing glucose production by the liver via gluconeogenesis [41]. This increase in gluconeogenesis is likely due to the increased levels of glucogenic amino acids in the blood, thus inducing glucose synthesis [42]. In regard to insulin resistance, some studies suggest that heightened amino acid levels in the blood intervene with insulin uptake via overactivation of the mTORC1 pathway, which increases phosphorylation of insulin receptor substrates and induces pancreatic beta cell dysfunction/degradation [28,30]. In turn, this phosphorylation and degradation could impair insulin-stimulated glucose uptake into cells [28,43]. Increased gluconeogenesis coupled with insulin resistance may be a potential explanation as to why the PG (increased protein intake) showed smaller improvements in fasting plasma glucose and insulin resistance [41]. Although both groups experienced improvements in insulin resistance, the positive effects of exercise on insulin resistance may have masked the full extent of negative impacts on insulin resistance from high protein consumption [34,44]. One randomized control trial (RCT) found similar results, specifically that a whey protein supplement did not enhance the effects of resistance training on glycemic control in overweight older adults with T2D [45]. Similarly, another RCT found no differences in glycemic control among a protein and control group—both of which underwent an exercise intervention [46]. Therefore, despite numerous findings of exercise interventions improving insulin resistance [19], a handful of studies—including the current study—suggest that protein may mitigate the insulin-sensitizing effects of physical activity [45,46].

Amino acids have been shown to directly reduce glucose uptake through the inhibition of insulin-induced glucose transport in skeletal muscle [41]. For instance, leucine supplementation may decrease muscular glucose uptake despite causing an increase in insulin levels [41,47]. Many studies highlight the impact that the ketogenic diet has on restoring insulin sensitivity in people with T2D and obesity [48,49]. This is highly relevant to the current study, as a ketogenic diet is not associated with high protein intake, but rather consists of high fat, moderate protein, and lower carbohydrate intake. Based on the negative effects of increased amino acid levels coupled with studies conducted on the ketogenic diet [41,48], moderate (rather than high) levels of protein intake may be a consideration to improve glycemic control [49]. If these findings are confirmed in larger randomized trials, this would have significant implications in the treatment of sarcopenic obesity where expert consensus suggests consumption of a high-protein diet [24].

The negative effects of branched-chain amino acid (BCAA) consumption on glucose control is also believed to have an impact on the mTORC1 pathway [50]. BCAAs activate mTORC1, which triggers a signaling pathway that ultimately phosphorylates IRS-1 [50]. Specifically, phosphorylation at specific sites of IRS-1 may cause substrate degradation and dampen insulin signaling [50]. This could further explain the higher fasting plasma glucose levels observed in the PG [50,51]. Increased blood glucose levels may also be associated with increased circulating inflammatory markers, and these biomarkers can further exacerbate insulin resistance [22]. This could potentially explain the smaller reductions in both fasting plasma glucose and inflammatory marker levels in the PG [52]. One RCT in older adults with obesity and T2D found similar results, as participants undergoing only a resistance training exercise intervention experienced significantly higher improvements in fasting glucose levels compared to those undergoing the same exercise intervention but with a protein supplement [45]. However, this same RCT showed significant improvements in circulating inflammatory markers among participants in the exercise plus protein intervention, which was not reproduced in the current study. Such heterogeneity in findings should prompt further research in hopes of reaching a more concrete conclusion about protein’s effects on both glucose homeostasis and inflammation.

This feasibility study did not demonstrate that protein intake had significant effects on inflammatory markers; however, the general trends observed were that the NPG had larger decreases in inflammatory markers. One diet-induced weight loss study in older adults found similar trends, as a low-protein diet (15% of caloric intake) had lower post-intervention levels of inflammatory markers compared to a high PG (30% of caloric intake) [53]. There is considerable heterogeneity across studies, as others have shown protein intake to be negatively associated with inflammatory markers [54]. Little evidence exists to establish a direct correlation between protein intake and inflammation, yet a common thread is that of weight loss—irrespective of protein intake—which results in a decrease in inflammatory markers such as CRP, IL-6, and TNF-α [54,55]. Thus, protein intake alone may not explain the results of this study, but larger decreases in body weight found in the NPG could be an underlying factor in the larger decreases in inflammatory markers observed in this group. Nevertheless, additional studies should strongly be considered to evaluate protein intake’s direct effects on inflammation while controlling for weight loss. Inflammation is critical to examine in the older adult population due to its tendency to accelerate aging and exacerbate pathogenesis [56,57]. These effects of inflammation can predispose the older population to various age-associated diseases such as obesity, arthritis, and cardiovascular disease [56].

These preliminary findings have considerable implications, as they dispute the typical evidence-based recommendations advised for older adults with obesity [58]. Older adults require a higher amount of protein than recommended for younger adults [59]. Few studies have examined the effects of protein intake in older adults with obesity while undergoing weight loss. Moreover, these findings are novel, given that they are based in a less-researched population and in a less-researched exercise–protein combined intervention. Specifically, this study found that protein supplementation may be associated with insulin resistance in older adults with obesity experiencing weight loss. Insulin resistance can then, in turn, promote higher levels of circulating insulin [60]. Concurrently, the nonsignificant decrease in fasting plasma glucose in the PG was also related to less of a change in inflammatory markers [52,61]. This is consistent with previous studies showing that macrophages exposed to raised plasma glucose increase mitogen-activated protein kinase (MAPK) signaling, which cause an increase in TNF-α secretion [52,62]. However, whether baseline inflammatory status impacts the metabolic adaptation observed with protein consumption or its utilization requires further evaluation. Furthermore, there were marked differences in waist circumference between groups that demonstrate some discrepancies with other anthropometric and body composition measures. Specifically, the PG showed a significantly larger decrease in waist circumference compared to the NPG despite showing smaller reductions in other measures such as weight change and fat mass index. Reductions in weight can lead to different changes in body composition—some participants losing more visceral fat (e.g., waist circumference) than global fat, for instance. These provocative findings merit further confirmation and/or evaluation on larger samples, as these findings may influence carbohydrate metabolism. While the primary goal of the parent study was feasibility and acceptability, this secondary analysis was not to provide more precise or nuanced results compared to previous studies, as this analysis was underpowered to make inferences. Importantly, these findings do prompt further research in related fields, but note that a future randomized controlled trial would be necessary to achieve these goals [46].

This study has several limitations that need to be acknowledged. First, this was a pilot, feasibility study and consisted of a relatively small cohort of 28 older adults. Second, participants enrolled in this pilot study were not randomly assigned into the PG and NPG, which heeds biases in unknown and baseline covariates (including smoking). Thus, these results suggest the need for future randomized controlled trials. Third, this was not a feeding study where all meals were controlled; thus, participants continued to have autonomy over their dietary intake with type and volume of food not being controlled for. However, each participant group consumed a minimum of 1.2 g/kg/day of protein—without factoring in the protein supplement. Fourth, it was also unclear whether timing of protein impacts weight loss, insulin resistance, or inflammatory markers, given that participants took one of the protein supplements at a time of their choosing. Fifth, it is unclear if the smaller reductions in inflammatory markers and lesser improvement in insulin resistance were due to reduced weight loss observed in the PG. Although protein supplementation may increase satiety, it nevertheless increases caloric intake if no changes are made to one’s overall diet and may result in less weight loss [8,63]. Lastly, these findings cannot be generalized from the protein or exercise intervention alone. There was a great deal of homogeneity in results. Specifically, both the PG and NPG experienced weight loss, as well as reductions in markers of insulin resistance and inflammation. Further, comparisons of these changes between groups were insignificant. Thus, it is unclear if these results are due to a single component of the intervention or the combined intervention. Future studies should be conducted that attempt to isolate protein’s effects on glucose homeostasis while undergoing weight loss in this population. Such studies could consist of factorial design or a sequential multiple assignment randomized control trial.

Given these promising effect sizes, future studies should be conducted with increased sample size in a heterogeneous population. To better isolate the effects of protein on glucose homeostasis with exercise, feeding studies also should be conducted to remove extraneous variables introduced through an autonomous diet [64]. It is also important to conduct studies in more detailed age groups—the oldest old population (≥80 years) [65]. Additional evaluation should focus on how different types of protein supplementation may have different effects on glucose homeostasis (soy versus whey protein, for instance) [41]. Furthermore, many studies have shown timing of caloric intake pre- vs. post-exercise to have heterogenous effects on muscle protein breakdown and synthesis, which could, in turn, also influence glucose homeostasis [34,66,67,68]. For instance, consuming a balanced meal following exercise may mitigate muscle protein breakdown early in the day [68], whereas others have found that ingesting protein every three hours was the best way to stimulate muscle protein synthesis [13,67]. Despite strides in nutrition research, little work has been conducted related to exercise–diet combined interventions; thus, more combined intervention studies, as in this pilot intervention, should be conducted.

## 5. Conclusions

Older adults with obesity pursuing weight loss demonstrated improvements in insulin resistance and inflammatory markers. However, this study found that consuming a whey protein supplement post-exercise may reduce such improvements. While the results were nonsignificant, effect sizes warrant a large, randomized trial with an adequate sample size to minimize potential confounding variables. Further, additional studies should be conducted in an attempt to replicate these findings in a larger cohort.

## Figures and Tables

**Table 1 nutrients-15-04947-t001:** Baseline characteristics. Display of participant age, sex, and sociodemographics.

	Protein Supplement (PG) (N = 14)	No Protein Supplement (NPG) (N = 14)	*p* Value
Age	72.9 (4.4)	73.0 (6.3)	0.94
Female sex	12 (85.7)	11 (78.6)	0.62
Marital status			0.07
Married	4 (28.6)	10 (71.4)	
Divorced/widowed	10 (71.4)	4 (28.5)	
Insurance status			
Medicare	14 (100.0)	14 (100.0)	1.00
Medicaid	0 (0.0)	0 (0.0)	1.00
Private insurance	8 (57.1)	10 (71.4)	0.44
Smoking status			0.005
Nonsmoker	6 (42.9)	13 (92.9)	
Former smoker	8 (57.1)	1 (7.1)	
Education			0.12
High school	2 (14.3)	0 (0.0)	
Some college	5 (35.7)	3 (21.4)	
College degree	3 (21.4)	5 (35.7)	
Post-college degree	4 (28.6)	6 (42.9)	
Drinks per week			0.54
None	6 (42.9)	7 (50.0)	
1 to 5	7 (50.0)	6 (42.9)	
6+	1 (7.1)	1 (7.1)	
Income			0.42
Less than USD 25,000	2 (14.3)	0 (0.0)	
USD 25,000 to USD 49,999	9 (64.3)	9 (64.3)	
USD 50,000 to USD 74,999	1 (7.1)	3 (21.4)	
USD 75,000 to USD 99,999	1 (7.1)	1 (7.1)	
USD 100,000+	1 (7.1)	1 (7.1)	
Comorbidities			
Anxiety	2 (14.3)	1 (7.1)	0.55
Coronary artery disease	1 (7.1)	2 (14.3)	0.55
Chronic obstructive pulmonary disease	1 (7.1)	0 (0.0)	0.32
Depression	3 (21.4)	3 (21.4)	1.00
Diabetes	3 (21.4)	2 (14.3)	0.63
Fibromyalgia	0 (0.0)	1 (7.1)	0.32
High cholesterol	5 (35.7)	4 (28.6)	0.69
Hypertension	7 (50.0)	7 (50.0)	1.00
Non skin cancer	1 (7.1)	0 (0.0)	0.32
Osteoarthritis	6 (42.9)	6 (42.9)	1.00
Rheumatologic disease	1 (7.1)	1 (7.1)	1.00
Sleep apnea	2 (14.3)	4 (28.6)	0.37
Stroke	0 (0.0)	0 (0.0)	1.00

Age is represented as mean (SD). All other metrics are represented as counts.

**Table 2 nutrients-15-04947-t002:** Changes in anthropometric and body composition measures. Shown are the chosen anthropometric and body composition variables and statistical analysis for the protein and nonprotein groups.

Variables	Protein Supplement	No Protein Supplement	*p*-Value	Effect Sizes
	(n = 14)	(n = 14)		
**Anthropometric Variables**				
% Weight change	−3.7 (3.1)	−6.0 (3.4)	0.07	0.79
Body mass index, kg/m^2^	−1.36 (1.09)	−2.15 (1.18)	0.08	0.71
Waist Circumference (cm)	−14.8 (45.5)	−6.45 (6.25)	0.03	0.88
**Body Composition Variables**				
Fat-free mass index,	−0.66 (0.63)	−0.23 (0.65)	0.08	0.68
Skeletal muscle mass, kg	−1.69 (2.51)	−1.67 (1.78)	0.98	0.01
Appendicular lean mass/BMI, m^2^	0.00 (0.01)	0.02 (0.01)	<0.001	0.30
Fat mass index	−0.97 (1.28)	−1.35 (0.91)	0.37	0.34
Visceral adipose tissue, L	−0.02 (0.81)	−1.02 (0.98)	<0.001	1.11

All values represented as mean (standard deviation). All values represent difference between pre- and post-intervention. BMI—body mass index.

**Table 3 nutrients-15-04947-t003:** **Markers of Glucose Homeostasis.** Shown are markers and statistical analysis used to evaluate glucose homeostasis within the protein and nonprotein groups.

Markers	Protein Supplement (n = 14)				No Protein Supplement (n = 14)					
	Baseline	Week 12	Difference	*p* Value	Baseline	Week 12	Difference	*p* Value	** Δ *p* Value	Effect Size
Fasting plasma glucose (mg/dL)	109.1 (21.9)	105.1 (17.3)	−4.0 (13.9)	0.30	113.1 (33.8)	100.9 (11.3)	−12.2 (25.8)	0.10	0.36	−0.39 (−1.14, 0.36)
Insulin levels (pmol/L)	109.8 (55.2)	102.0 (61.2)	−7.8 (28.2)	0.31	119,4 (89.4)	72.6 (33.0)	−46.8 (60.6)	0.01	0.04	−0.82 (−1.58, −0.04)
C-peptide (ng/mL)	3.01 (1.1)	2.99 (1.36)	−0.02 (0.67)	0.91	3.58 (1.98)	2.89 (1.03)	−0.69 (1.25)	0.06	0.17	−0.66 (−1.42, 0.11)
Insulin resistance (HOMA-IR)	2.43 (1.19)	2.25 (1.35)	−0.18 (0.64)	0.32	2.68 (2.16)	1.60 (0.75)	−1.08 (1.50)	0.02	0.05	−0.78 (−1.54, −0.001)
% β-cell function	119.7 (41.6)	115.8 (36.3)	−3.9 (22.6)	0.52	117.7 (37.4)	100.9 (21.4)	−16.8 (33.9)	0.09	0.27	0.45 (−1.19, 0.31)

** Adjusted for age, sex, weight loss. All values represented are mean (SD) or effect size (95% CI). Δ*p* value: *p* value for the differences (baseline/follow-up) column, between groups.

**Table 4 nutrients-15-04947-t004:** **Markers of inflammation.** Inflammatory markers and their statistical analysis used to evaluate inflammatory status within the protein and nonprotein groups.

	Protein Supplement (n = 14)				No Protein Supplement (n = 14)					
Markers	Baseline	Week 12	Difference	*p* Value	Baseline	Week 12	Difference	*p* Value	** Δ *p* Value	Effect Size
CRP (mg/L)	5.01 (4.19)	4.36 (3.64)	−0.65 (1.86)	0.21	5.09 (4.02)	3.09 (2.18)	−2.00 (3.96)	0.08	0.16	−0.44 (−1.18, 0.32)
ESR (mm/h)	13.4 (4.5)	12.4 (3.7)	−0.93 (1.82)	0.08	17.0 (11.4)	14.9 (9.2)	−2.14 (5.30)	0.16	0.19	−0.31 (−1.05, 0.44)
TNF-α (pg/mL)	1.16 (0.37)	1.09 (0.32)	−0.08 (0.28)	0.33	1.30 (0.43)	1.08 (0.43)	−0.19 (0.23)	0.01	0.14	−0.45 (−1.21, 0.32)
IL-6 (pg/mL)	2.73 (1.38)	2.29 (1.12)	−0.44 (1.26)	0.21	2.68 (1.93)	2.92 (2.51)	0.10 (2.14)	0.87	0.18	0.31 (−0.47, 1.09)
* IL-1β				<0.001				<0.001	0.13	
<5 (pg/mL)	13 (92.9%)	13 (92.9%)	0 (0.0%)		9 (64.3%)	11 (78.6%)	0 (0.0%)			
5–50 (pg/mL)	1 (7.1%)	1 (7.1%)	0 (0.0%)		3 (21.4%)	2 (14.3%)	2 (66.7%)			
>50 (pg/mL)	0 (0.0%)	0 (0.00%)	---		2 (14.3%)	1 (7.1%)	1 (50.0%)			

All values represented are mean (SD), counts (%), or effect size (95% CI). ** Adjusted for age, sex, weight loss. Δ*p* value: *p* value for the differences (baseline/follow-up) column, between groups. CRP = C-reactive protein; ESR = erythrocyte sedimentation rate; TNF-α = tumor necrosis factor-alpha; IL-6 = interleukin-6; IL-1β = interleukin-1 beta. * IL-1β is expressed as count (%) for the number of participants with IL-1β levels <5, 5–50, and >50 pg/mL, respectively.

## Data Availability

Data is available upon request and institutional data use agreements.

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
