# Peer review of "Protein Supplementation May Dampen Positive Effects of Exercise on Glucose Homeostasis: A Pilot Weight Loss Intervention"

_nutrients, 2023, doi:10.3390/nu15234947_

Round 1
Reviewer 1 Report
Comments and Suggestions for Authors
Comments on nutrients--2712279-peer-review-v1
Major comments:
This study is based on the negative impact of Protein Supplementation on Glucose Homeostasis in Older Adults with increased body weight. The literature review is not strong enough to detect research gaps.. This study has several limitations. To improve the significance of the work the number of respondents should be increased. In addition, participants were counselled, and they were not monitored. Many of them continued to have autonomy in dietary intake with the type and volume of food. There are no figures in the manuscript, so the authors are advised to add several figures and present the data graphically. The figures should be attractive and eye-catching. The novelty of this work is not clearly highlighted. The Results part of the manuscript is confusing for readers and needs more clarity and better construction. The discussion of the results is very poor. Authors are advised to compare the results with the published literature and discuss results critically with the previous studies. The conclusion section is missing completely. The number of references is not enough. The authors are advised to revise and upgrade references, including the latest references. It is evident that the quality of the manuscript does not meet the standards of the Journal and should be considered for major revision.
Introduction:
Literature survey is poor and does not meet standards in writing a research paper. It is not clear how this study gives more precise results than previous studies. It needs to be strengthened in terms of recent research in this area with possible research gaps. The authors are also advised to emphasize the novelty of the study. The introduction needs to emphasize the research work with a detailed explanation of the whole process considering past, present, and future scope. Research gaps should be highlighted more clearly and future applications of this study should be added. More issues need to be addressed, see specific comments.
Specific comments:
1. Page numbering has no traceability.
2. Keywords are missing. The authors are advised to add them.
3. Abstract: Several parameters are missing in the abstract section such as clearly mentioned objectives, a novelty that is insufficient to delineate the whole picture of contribution, keywords and possible outcomes and application of this study. Also, it is recommended that the authors shorten it. Hence, it is suggested to add these points to make it a standard abstract (250 words maximum). Some sentences need more clarity and better construction.
4. The introduction section is not strong enough to provide research gaps. Authors are advised to discuss some latest references such as given below: doi:10.1001/jamainternmed.2020.4153; doi: 10.17392/1279-21, doi.org/10.3390/nu12010222; doi.org/10.1007/s12033-023-00813-z.
5. Page 1, line 15: “While higher dietary protein intake has been associated with an increased risk of type 2 diabetes, losing weight improves glucose metabolism and homeostasis. While some stud-1ies have also highlighted protein as a key to improving glucose homeostasis, other studies have found the opposite to be true highlighting the heterogeneity in treatments.” The sentences need better construction. The authors advised to restructure them.
6. Page 1, line 18: “We evaluated whether a multicomponent weight loss intervention augmented with protein supplementation in older persons aged 65+ years with obesity impacts glucose homeostasis.” The Authors are advised to rewrite the paper in passive voice. Please carefully go through the entire manuscript and correct this. It’s better to say “In this paper, the influence of a multicomponent weight loss intervention augmented with protein supplementation in older persons aged 65+ with obesity impacts glucose homeostasis was investigated.”
7. Page 1, line 22: “A 12-week, non-randomized, parallel group intervention of protein (PG) and non-protein (NPG) arms for 28 older rural adults (body mass index (BMI) ≥30 kg/m2) was conducted at a community ageing center.” The authors are advised to pay attention to subscripts and superscripts as they completely change the meaning (move the number 2 into an exponent (kg/m2). Please go through the entire manuscript and correct this.
8. Page 1, line 24: “The PG was provided 24 with whey protein supplementation three times weekly post-strength training. Body composition 25 was assessed using a Seca mBCA 514.” The authors are advised to write the full name of the abbreviation on the first mention (medical Body Composition Analyzer). Please check the abbreviations in the whole manuscript.
9. Page 1, line 34: “There was smaller decrease in pre- vs. post fasting glucose levels (PG:-4mg±13.9 vs. NPG: -12.2±25.8 mg/dL, effect size (ES): -0.40; p=0.10), insulin (-1.32±4.68 vs. -7.79±10.14 mclU/mL, ES: 0.82 ;p=0.01), and c-peptide (-0.02±0.67 vs. -0.69±1.25 ng/mL, ES: 0.66; p=0.06) in the PG at follow-up. HOMA-IR (-0.18±0.64 vs. -1.08±1.50, ES: 0.78; p=0.02) and percent β-cell function (-3.94±22.61 vs. -16.81±33.85%; 36 ES: 0.45; p=0.09]) also decreased less in the PG” The authors are advised to use standard units (IUPAC system) to express insulin levels (mU/L or pmol/L).
10. Page 1, line 48: “Insulin resistance—a disruption in glucose homeostasis—is strongly associated with the development of T2D, abdominal obesity, and key components of the metabolic syndrome, all of which have a tendency to be pro-inflammatory”. A comma before “and” is redundant, authors are advised to remove it.
11. Avoid lumping references. Instead, summarise the main contribution of each referenced paper in a separate sentence. Please carefully go through the entire manuscript and correct this.
12. Page 2, line 66: “Specifically, studies suggest that ingesting a protein bolus immediately after resistance exercise upregulates muscle protein synthesis (MPS) in older adults through mTORC1 pathway activation” What is mTORC1? The authors are advised to write the full name of the abbreviation on first mention.
13. Page 2, line 78: “To begin answering these questions, we conducted a weight loss intervention in older adults with obesity who were non-randomly provided a whey protein supplement (protein group, PG) or not (non-protein group, NPG) to evaluate its impact on glucose homeostasis.” The Authors are advised to remove all personal pronouns. Please carefully go through the entire manuscript and correct this.
14. Authors are advised to revise the titles in the manuscript.
15. Page 3, line 136: “Mean age was 72.9±4.4 and 73.0± 6.3 years in the PG and NPG respectively (p=0.94) 136 (Table 1).” The sentence is confusing for readers and needs more clarity and better construction. It’s better to say the mean age was 72.9±4.4 in the PG and 73.0± 6.3 years in the NPG respectively (p=0.94) (Table 1).
16. Page 3, lines 136-114: The Results part of the Manuscript is confusing for readers and needs more clarity and better construction.
17. Page 4, line 155: “Table 1. Baseline Characteristics”: What role does marital status play in research?
18. Page X, line 190: “Our preliminary findings align with previously conducted studies.” The authors didn’t specify which research it was about. Give references of previous research.
19. The conclusion section is missing completely. Authors are advised to add a conclusion (up to 250 words) with the main results, contribution and limitations of the study.
20. The number of references is not enough. The authors are advised to revise and upgrade references, including the given latest references.
Comments on the Quality of English Language
Minor editing required.
Reviewer 2 Report
Comments and Suggestions for Authors
The present manuscript presents a pilot study aiming to evaluate the effects of a whey protein supplement administration is older adults, on glucose homeostasis.
The topic is interesting, but the research has serious flaws.
Abstract - the authors specify that the decrease in insulin resistance was not significant, therefore it is a contradiction to declare that the consumption of protein had beneficial effect on glucose homeostasis. Also, this is in contradiction with the articles' title.
Why were the participants allocated non-randomly in groups? There is a significant difference between the two groups with regard to the non-smoking status, therefore, since smoking can impact the results, I consider that the study’s results could be biased.
Also, since the interventions included protein administration and exercises, it is complicate to differentiate the effects of protein supplementation by the effects produces by exercise. Two more groups (protein group and non-protein group without exercise) should have been included in the present study for consistent results.
More details about the type of proteins administered are required.
Comments on the Quality of English LanguageMinor editing of English language required
Round 2
Reviewer 1 Report
Comments and Suggestions for Authors
Paper is improved.
Comments on the Quality of English LanguageModerate editing required.
Author Response
Thank you for the comment. The authors have made one change to the title of Table 2 which now reads as follows: "Table 2. Changes in Anthropometric and Body Composition Measures."
If there are any specific edits to language that the reviewer would like, the authors would be happy to make this change.
Reviewer 2 Report
Comments and Suggestions for Authors
The manuscript has been improved, and I understood the authors opinion regarding the present study.
Author Response
Thank you for the comment. The authors made one change to the title of Table 2, which now reads as follows:
"Table 2. Changes in Anthropometric and Body Composition Measures."